# Study protocol for the epigenetic characterization of angor pectoris according to the affected coronary compartment: Global and comprehensive assessment of the relationship between invasive coronary physiology and microRNAs

Lucía Matute-Blanco[1][☯], Diego Fernández-Rodríguez[1][☯]*, Juan Casanova-Sandoval[1], Thalía Belmonte[2,3], Iván D. Benítez[2,3], Kristian Rivera[1], Marcos Garcia-Guimaraes[1], Carlos Cortés Villar[4], Vicente Peral Disdier[5], Raúl Millán Segovia[5], Ignacio Barriuso[1], David de Gonzalo-Calvo[2,3], Ferran Barbé[2,3], Fernando Worner[1]

1 Department of Cardiology, Institut de Reçerca Biomèdica de Lleida (IRBLleida), University Hospital Arnau de Vilanova, Lleida, Spain, 2 Institut de Reçerca Biomèdica de Lleida (IRBLleida), Translational Research in Respiratory Medicine Group, Lleida, Spain, 3 Institute of Health Carlos III, CIBER of Respiratory Diseases (CIBERES), Madrid, Spain, 4 Department of Cardiology, University Hospital Miguel Servet, Zaragoza, Spain, 5 Department of Cardiology, University Hospital Son Espases, Palma de Mallorca, Spain

☯ These authors contributed equally to this work.

* dfernandez.lleida.ics@gencat.cat

## Abstract

### Background

MicroRNAs (miRNAs) are noncoding RNAs involved in post-transcriptional genetic regulation with a proposed role in intercellular communication. miRNAs are considered promising biomarkers in ischemic heart disease. Invasive physiological evaluation allows a precise assessment of each affected coronary compartment. Although some studies have associated the expression of circulating miRNAs with invasive physiological indexes, their global relationship with coronary compartments has not been assessed. Here, we will evaluate circulating miRNAs profiles according to the coronary pattern of the vascular compartment affectation.

### Study and design

This is an investigator-initiated, multicentre, descriptive study to be conducted at three centres in Spain (NCT05374694). The study will include one hundred consecutive patients older than 18 years with chest pain of presumed coronary cause undergoing invasive physiological evaluation, including fractional flow reserve (FFR) and index of microvascular resistance (IMR). Patients will be initially classified into four groups, according to FFR and IMR: macrovascular and microvascular affectation (FFR$\leq$0.80 / IMR$\geq$25), isolated macrovascular affectation (FFR$\leq$0.80 / IMR<25), isolated microvascular affectation (FFR>0.80 / IMR $\geq$25) and normal coronary indexes (FFR>0.80 / IMR<25). Patients with isolated

**Data Availability Statement:** No datasets were generated or analysed during the current study. All relevant data from this study will be made available upon study completion.

**Funding:** DdGC has received financial support from Instituto de Salud Carlos III (Miguel Servet 2020: CP20/00041) co-funded by Fondo Social Europeo Plus (FSE+). CIBERES is an initiative of the Instituto de Salud Carlos III.

**Competing interests:** The authors have declared that no competing interests exist.

microvascular affectation or normal indexes will also undergo the acetylcholine test and may be reclassified as a fifth group in the presence of spasm. A panel of miRNAs previously associated with molecular mechanisms linked to chronic coronary syndrome will be analysed using RT-qPCR.

## Conclusions

The results of this study will identify miRNA profiles associated with patterns of coronary affectation and will contribute to a better understanding of the mechanistic pathways of coronary pathology.

## Introduction

In the coronary artery circulation, there is a progressive branching of the vessels, which conform to two compartments that can be diseased individually or in combination: a) the macrovascular compartment: epicardial arteries (conduction function); and b) the microvascular compartment: arterioles (flow regulating function) and capillaries (exchange function).

Invasive physiological coronary indices allow for a thorough examination of macro and microvascular compartments, allowing one to precisely discern their involvement [1–4]. Although coronary compartments have traditionally been evaluated separately, evidence suggests that a global evaluation of coronary circulation improves management of patients with ischemic heart disease.

MicroRNAs (miRNAs) comprise a group of small noncoding RNA transcripts that regulate gene expression at the post-transcriptional level. miRNAs play a critical role in cellular pathways involved in physiology and pathology [5], including ischemic heart disease [6]. Although miRNAs were initially identified as intracellular regulators, they have also been described in different body fluids. In this scenario, circulating miRNAs have emerged as potential biomarkers with clinical application in medical decision-making [7]. A role as endocrine genetic signals has also been proposed [6,8]. To date, the role of miRNAs in coronary function has been independently evaluated for macro and microcirculation [9,10]. However, studies assessing the association between circulating miRNAs and the global coronary pattern are currently lacking [8].

Therefore, the objective of the current investigation will be to evaluate globally and comprehensively the relationship between different patterns of coronary involvement, using invasive methods, and circulating miRNA levels.

## Materials and methods

### Study design

Multicentre descriptive study to be conducted in three centres in Spain. This study is investigator-initiated, promoted by the Institut de Reçerca Biomèdica de Lleida (IRBLleida). The list of participant centres and investigators are detailed in the **S1 File**.

This study is adhered to the principles outlined in the Declaration of Helsinki. The approval was granted by the 'Comité de Ética de Investigación con Medicamentos del Hospital Arnau de Vilanova de Lleida´ (Ethics Committee for Drug Research of the Arnau de Vilanova of Lleida) with the CEIC-2665 registry, on March 31st, 2022. The study is registered at Clinical-Trials.gov (NCT number: NCT05374694).

| | STUDY PERIOD | | | | | | | |
| --- | --- | --- | --- | --- | --- | --- | --- | --- |
| | Enrolment | Allocation | Post-allocation | | | | | Close-out |
| TIMEPOINT** | $-t_1$ | 0 | $t_1$ | $t_2$ | $t_3$ | $t_4$ | etc. | $t_x$ |
| **ENROLMENT:** | | | | | | | | |
| **Eligibility screen** | X | - | | | | | | |
| **Informed consent** | X | - | | | | | | |
| *[List other procedures]* | X | - | | | | | | |
| **Allocation** | N/A | | | | | | | |
| **INTERVENTIONS:** | | | | | | | | |
| *[Intervention A]* | N/A | | | | | | | |
| *[Intervention B]* | N/A | | | | | | | |
| *[List other study groups]* | N/A | | | | | | | |
| **ASSESSMENTS:** | | | | | | | | |
| *[List baseline variables]* | X | - | | | | | | |
| *[List outcome variables]* | X | - | | | | | | |
| *[List other data variables]* | X | - | | | | | | |

*Recommended content can be displayed using various schematic formats. See SPIRIT 2013 Explanation and Elaboration for examples from protocols.
**List specific timepoints in this row.

**Fig 1. Schedule of enrolment, interventions and assessments.**

The hypothesis to be tested is whether different patterns of coronary affectation are associated with the expression of certain miRNAs. To evaluate coronary patterns, invasive physiological evaluation, including "fractional flow reserve" (FFR) [1,11,12] and "index of microcirculatory resistance" (IMR) [1,13], a coronary guide pressure-temperature will be used.

The schedule of enrolment, interventions and assessments and the timeline and flowchart of the study are detailed in **Figs 1** and **2**.

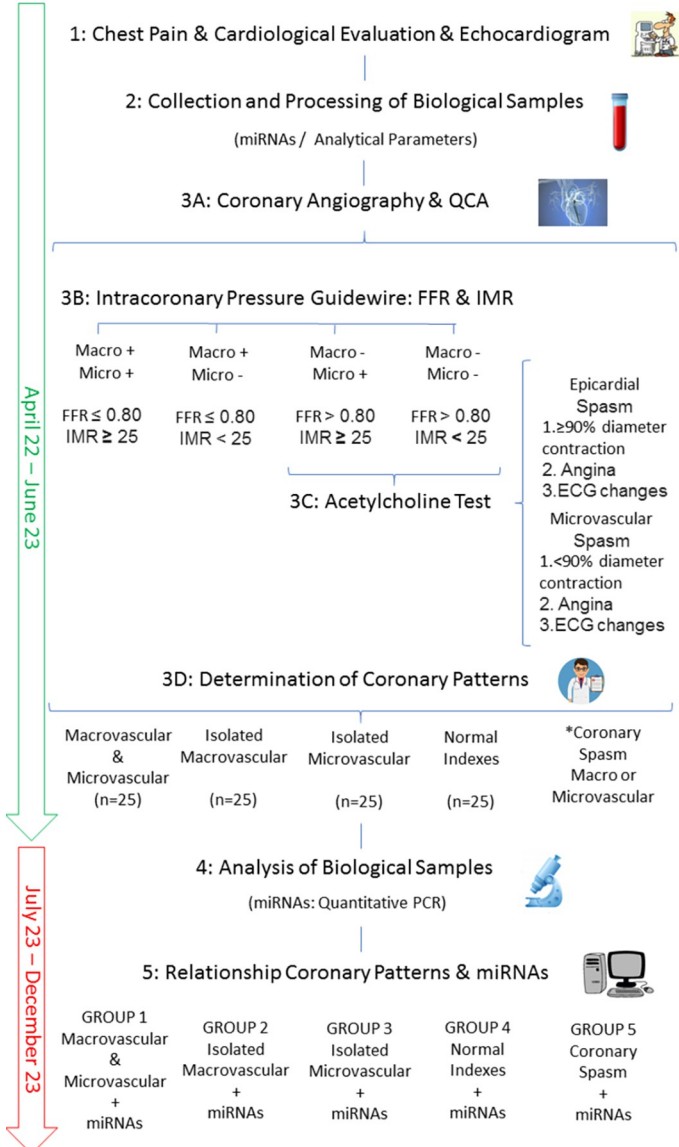

**Fig 2. Timeline and flowchart of the study.** Inclusion of the patients will be since April 2022 to June 2023. Analysis of biological samples and statistical analysis will be performed since July 2023 to December 2023. miRNAs, microRNAs; QCA, quantitative coronary angiography; FFR, fractional flow reserve; IMR, index of microcirculatory resistance; ECG, electrocardiographic.

## Patient selection

This study will include patients with suspected angina referred for coronary angiography and eventual angioplasty, accordingly to the recommendations of the European Society of Cardiology [1,14]. Inclusion and exclusion criteria are detailed in **Table 1**. It is noteworthy that if a total occlusion of any coronary artery is detected on coronary angiography, precluding measurements with pressure-temperature guidewires; the patient will be excluded for the initial purpose of the study and the biological samples for miRNAs analyses discarded.

**Table 1. Inclusion and exclusion criteria.**

| Inclusion criteria | Exclusion criteria |
|---|---|
| 1. Age $\geq$ 18 years. <br> 2. Patients with chest pain suggestive of angina evaluated by a cardiologist and referred for coronary angiography and eventual coronary angioplasty. <br> 3. Echocardiogram to rule out non-coronary cardiac causes of chest pain. <br> 4. Informed consent. | 1. Contrast allergy not susceptible to premedication. <br> 2. Severe bronchial asthma or adenosine intolerance. <br> 3. Atrio-ventricular block ($\geq 2^{nd}$ degree) or acetylcholine intolerance. <br> 4. Acute myocardial infarction with ST-segment elevation. <br> 5. Acute myocardial infarction without ST-segment elevation. <br> 6. Cardiogenic shock. <br> 7. Total occlusion of any coronary artery that precludes measurements with pressure-temperature guidewires. <br> 8. Previous coronary artery bypass grafting. <br> 9. Women with the possibility of being pregnant. <br> 10. Renal dysfunction with an estimated glomerular filtration rate <30 mL/min/1.73m2. <br> 11. Inability to understand the nature of the study and / or sign informed consent. <br> 12. Any other medical condition that, in the opinion of the researcher, may lead to safety issues for patients or may alter the results of the study. |

## Invasive physiological evaluation of the coronary circulation: Determination of coronary patterns

After diagnostic coronary angiography, a standardized analysis of the coronary lesions in terms of percentage of stenosis, lesion length, and estimated diameter of the vessel will be performed by quantitative coronary angiography (QCA).

Invasive physiological measurements will be performed according to recommendations [1,11–16]. The left anterior descending coronary artery will be evaluated in all patients and the left circumflex or right coronary artery in the event of a stenosis $\geq$ 30% by QCA, being possible to interrogate more than one coronary artery in the same patient. Furthermore, in patients referred for coronary angiography who have a previous positive ischemia test, the artery corresponding to the compatible territory with the detected ischemia will also be evaluated. An intracoronary pressure and temperature sensor-tipped guidewire (Pressure Wire ™ X guidewire 0.014', Abbott, IL, USA) will be used to perform the measures. The tip pressure sensor will be advanced to the mid-to-distal portion of the vessel to investigate. Baseline aortic pressure (Pa) and distal intracoronary pressure (Pd) will be obtained and resting indexes Pd/Pa and "resting full-cycle ratio" (RFR) will be determined. To measure the mean transit time (Tmn) under basal conditions, intracoronary administration of 3 mL of room-temperature saline will be injected three times manually in succession (3 mL/s). Then maximal hyperemia will be induced using adenosine iv (140 to 180 mg/kg/min) as a vasodilator drug and three additional intracoronary room temperature saline boluses of 3 ml will be administered to determine the maximum coronary flow presented as peak Tmn. Finally, the FFR, the "coronary flow reserve" (CFR) and the IMR will be calculated, using the software Coroventis Coroflow (Coroventis AB, Uppsala, Sweden). The calculation and pathological cut-off values of FFR ($\leq$ 0.80) and IMR ($\geq$ 25) are described in **Table 2** [1,11–14].

For the coronary vasospasm test, acetylcholine will be used [16]. Due to the risk of epicardial spasm in case of macrovascular ischemia, the acetylcholine test will be only performed in patients with nonpathological FFR values (>0.80). More detailed recommendations for the performance of coronary physiology studies are shown in **S2 File** [1,11–16].

**Table 2. Calculation and pathological values of FFR and IMR.**

| FFR = Pd / Pa | IMR = Pd × Tmn |
|---|---|
| FFR: fractional flow reserve | IMR: index of microcirculatory resistance |
| Pd: distal coronary pressure | Pd: distal coronary pressure |
| Pa: aortic pressure | Tmn: mean transit time |
| Pathological value: FFR ≤ 0.80 | Pathological value: IMR ≥ 25 |

*Both FFR and IMR are calculated at maximal hyperemia.

Once the parameters have been obtained, the decision of possible medical, percutaneous, or surgical treatment of epicardial coronary lesions leading to ischemia will be made based on current clinical practice guidelines recommendations [14].

The coronary pattern corresponding to each patient will be determined based on FFR and IMR values. Macrovascular affectation will be defined whether the FFR value is ≤0.80 and microvascular affectation whether the IMR value is ≥25 [1,11–14]. Different groups related to coronary patterns are shown in the **Table 3**. Since the present protocol also contemplates an approach to the study of coronary spasm, an additional group will be considered, through the performance of the acetylcholine test in patients with isolated microvascular affectation or normal coronary indices. In the presence of macro or microvascular spasm, patients will be included in *Group 5* [16].

## Sample collection: MicroRNAs and analytical parameters

In each patient, before coronary angiography and any drug administration; 6 mL of blood will be drawn for the centralized miRNAs analyses. Samples will be obtained with support by IRBLleida Biobank (B.0000682) and Plataforma Biobancos PT20/00021. Blood samples will be collected in ethylenediaminetetraacetic acid (BD) blood collection tubes by venipuncture and centrifuged to separate plasma (1500 × g for 10 minutes). Plasma supernatant will be immediately aliquoted, frozen, and stored at -80˚C until miRNA profiling. The analysis of other biochemical and/or haematological parameters (**S3 File**) will be performed in the laboratories of each participating centre.

**Table 3. Study groups according to invasive physiological evaluation.**

| FFR & IMR | | |
|---|---|---|
| Group 1 | Macrovascular & Microvascular | FFR ≤ 0.8 IMR ≥ 25 |
| Group 2 | Isolated Macrovascular | FFR ≤ 0.8 IMR < 25 |
| Group 3 | Isolated Microvascular | FFR > 0.8 IMR ≥ 25 |
| Group 4 | Normal indexes | FFR > 0.8 IMR < 25 |
| **Acetylcholine Test** | | |
| Group 5* | Epicardial spasm OR Microvascular spasm | ≥90% diameter contraction Chest pain ECG changes <90% diameter contraction Chest pain ECG changes |

FFR, fractional flow reserve; IMR, index of microvascular resistance; ECG, electrocardiographic.

*Only patients in groups 3 and 4 will undergo the acetylcholine test.

## microRNA profiling

miRNA profiling will be performed in the laboratory of the TRRM group of the IRBLleida by experienced personnel without access to clinical data. All experiments will be performed using the gold-standard technique; RT-qPCR, under standardized conditions in the same laboratory and according to the previous methodology used by the research group [17,18].

Total RNA will be isolated from 200μL of frozen plasma samples using the miRNeasy Serum/Plasma Advanced Kit (Qiagen), according to the manufacturer's instructions. Synthetic *Caenorhabditis elegans* miR-39-3p (cel-miR-39-3p) will be added as an external reference RNA ($1.6 \times 10^8$ copies/μL) to normalize differences in the RNA isolation efficiency. The mixture will be supplemented with 1 μg of carrier RNA from bacteriophage MS2 (Roche) to improve RNA yield and the RNA spike-in Kit (Qiagen) for quality control. RNA purification will be performed with RNeasy UCP MinElute spin columns according to the manufacturer's recommendations. RNA will be stored in a -80°C freezer until further analysis.

miRNA profiling will be performed according to the protocol of the miRCURY LNA Universal RT microRNA PCR system (Qiagen). RT reactions will be performed using the miRCURY LNA RT Kit (Qiagen) in a total volume of 10 μL under the following conditions: incubation for 60 minutes at 42°C, inactivation for 5 minutes at 95°C and immediate cooling to 4°C. cDNA will be stored at −20°C. Individual miRNAs will be analysed using miRCURY LNA miRNA Custom Panels (384-well plates) (Qiagen). qPCR will be carried out in a QuantStudio[TM] 7 Flex Real-Time PCR System (Applied Biosystems, Waltham, MA, USA) in a total volume of 10 μL. The qPCR conditions will be 95°C for 2 minutes, followed by 40 cycles of 95°C for 10 seconds and 5°C for 1 minute, followed by analysis of the melting curves. UniSp3 Interplate Calibrator will be analysed to account for variability between the plates. Negative controls excluding the template from the RT and PCR and the enzyme mix in the RT reaction will be included. The amplification curves will be evaluated using QuantStudio Software v1.3 (Thermo Fisher Scientific, Massachusetts, USA). The quantification cycle (Cq) above 35 cycles will be considered undetectable and censored at the minimum level observed for each miRNA. miRNAs in which 80% of the samples meet these criteria will be considered to be below the limit of detection. Relative quantification will be performed using the $2^{-dCq}$ method, where $\Delta Cq = Cq_{miRNA} - Cq_{cel-miR-39-3p}$. Expression levels will be logarithmically transformed for statistical purposes. Global mean expression for each sample will be compared between independent centres in order to analyse potential confounding.

A panel of 14 circulating miRNAs has been selected after an extensive review of the literature [19–29]. The panel included miRNAs described as biomarkers of chronic coronary syndrome and previously associated with molecular pathways that could be altered in chronic coronary syndrome in *in vitro*, *in vivo*, and patient-based approaches (**Fig 3**). Other potential miRNAs related to the involvement of the macro and/or microvascular coronary compartment may be considered.

## Molecular pathway and gene ontology analyses

Bioinformatic prediction analysis will be performed using the web-based computational tool DIANA-miRPath v3.0 [30]. DIANA-miRPath v3.0 combines information on manually curated experimentally validated miRNA: gene interactions from TaRBase v7.0 with the Kyoto Encyclopedia of Genes and Genomes (KEGG) database and Gene Ontology (GO) annotations (for biological processes). Enrichment analysis will be performed using Fisher's exact test (hypergeometric distribution). The adjusted p-value for false discovery rate (FDR) will be set at <0.05.

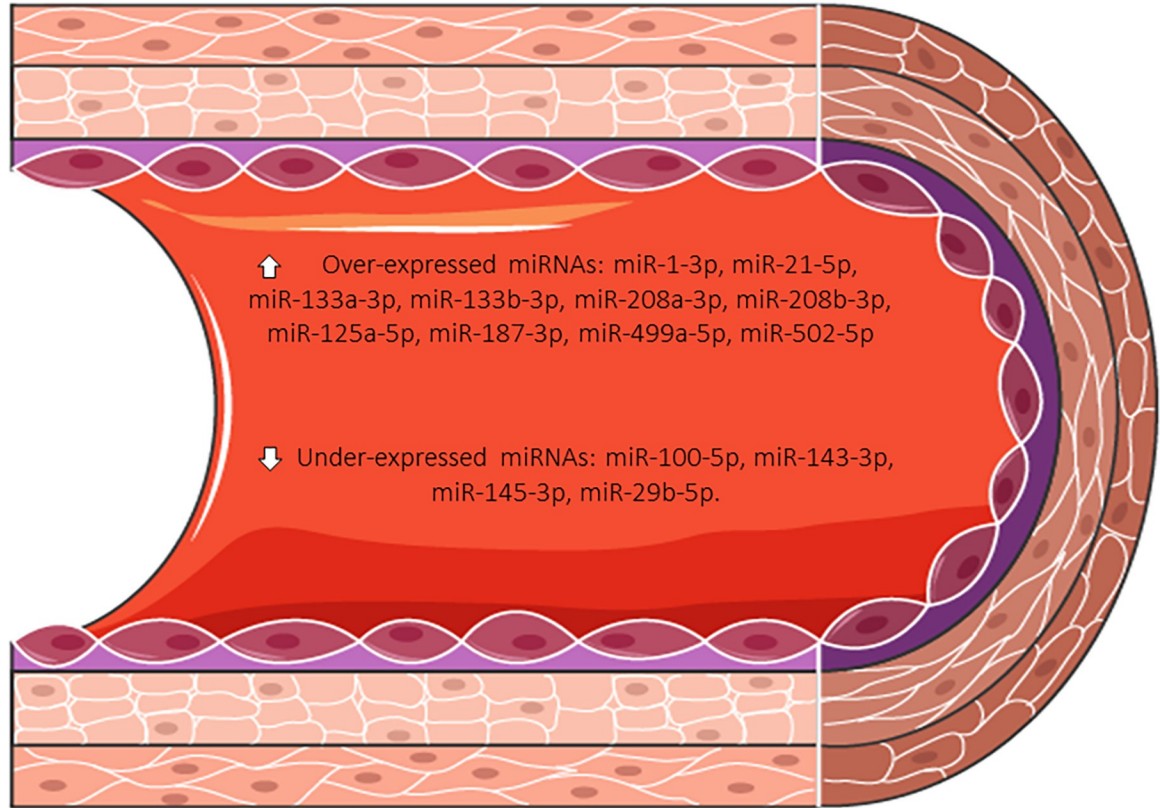

**Fig 3. miRNAs to be evaluated in the study.**

## Data management and plan to share individual participant data

Each centre will fill an anonymized and predefined case report form (CRF) developed by the investigators. The selected variables are oriented to the cardiovascular risk factors, comorbidities, clinical findings, ischemia detection tests, medications, analytical parameters, echocardiographic and angiographic findings, and invasive physiological indexes (**S3 File**). All data collected during the study, after deidentification, will be shared. Data obtained through this study may be provided to qualified researchers with academic interest in cardiovascular diseases. Approval of the request and execution of all applicable agreements are prerequisites for the sharing of data with the requesting party.

## Sample size estimation

Because studies evaluating the association between circulating miRNAs and coronary patterns, assessed by invasive physiological methods, are currently lacking, the sample size was not calculated using standard procedures. Patients of Group *4* (normal coronary indexes) will be used as control group to the *Groups 1 to 3* until the recruitment of 25 patients per group will be completed, in a similar number to studies that evaluated the relationship of miRNAs with coronary endothelial function in patients undergoing cardiac catheterization [10,28,29], for a total sample of 100 patients. Based on the t test for the differential expression of miRNAs between two groups, accepting an alpha risk of 0.05 in a bilateral test with 25 patients per group, there is a statistical power of 95% to detect a difference of one standard deviation as statistically significant. It should be noted that patients in *Group 5* (coronary spasm) may overlap

with patients in *Groups 3 and 4*, therefore, *Group 5* has not been considered for sample size estimation.

## Statistical analysis plan

The characteristics of the study population will be summarized through standard descriptive statistics. Differences in variables between study groups and correlations between continuous variables will be analysed using the appropriate test (parametric or nonparametric). Differential expression of the miRNAs between groups will be evaluated using linear models with empirical Bayes statistics [31]. Unadjusted and adjusted models will be fitted, considering a propensity score (PS) to adjust by confounding factors. The PS will be defined as the probability of belonging to a given study group and it will be estimated using a logistic regression model that includes cardiovascular risk factors, medical conditions and medications as predictor variables. Unsupervised clustering of patients based on miRNA levels will be performed using parsimonious Gaussian mixture models [32]. The number of latent profiles will be determined using the parsimony criteria based on the minimum value of the Bayesian information criterion measure from 0 to 5 latent profiles. Each patient will be assigned to one class according to his or her highest computed probability of membership. The relationship between the latent profiles and study groups will be evaluated using Fisher´s exact test. In addition, we will use a machine learning process based on random forest to construct miRNAs signatures associated with discrimination of study groups. Variable importance plots will be displayed to illustrate the prediction value of each miRNA. The statistical software package R (www.r-project. org) will be used for statistical analyses. The assumptions of the statistical models used in the analyses will be adequately tested. The two-tailed significance level will be set at <0.05.

## Discussion

In this multicentre observational study, we aim to investigate the association between circulating miRNA levels and the affectation of coronary compartments globally. The main focus of this study will be on the coronary circulatory function regulated by miRNAs.

## Comprehensive physiological assessment of coronary circulation: Role of pressure and flow-derived indexes

Currently, invasive physiological evaluation has become an essential tool to guide myocardial revascularization, characterize endotypes, and discriminate patients at high risk of adverse cardiovascular outcomes in scenarios of chronic coronary syndrome [1,14].

The direct relationship between coronary pressure gradients and the coronary blood flow at maximal vasodilation of resistance vessels is the physiological rationale for the development of pressure and flow-derived indexes. Thus, knowing the pressure gradient and the blood flow through the coronary compartments, it is possible to determine the circuit resistances and accurately characterize the function of the coronary circulation [2,3].

Studies mainly conducted by Pijls and De Bruyne [11,12] served to establish physiological bases of FFR and determine the expected cut-off points for this pressure-derived index. The authors were able to establish FFR as the "gold standard" for the detection of myocardial ischemia related to epicardial coronary stenose by a prospective validation, integrating information from different ischemia detection tests through a prospective multi-testing Bayesian approach. Since then, invasive coronary indexes have been considered the best tool for detecting myocardial ischemia. Later, Fearon et al. [13], by relating the pressure and the blood flow across microcirculation, established IMR as a flow-derived index to assess microcirculatory function.

Although there are alternative indexes that estimate coronary flow, by assessing flow velocities with Doppler-equipped guidewires, the Doppler signal is not analysable in up to 30% of patients [33], which could make it difficult to classify patients into study groups.

In our study protocol, we selected FFR and IMR to perform an exhaustive evaluation of both coronary compartments to clearly define the possible patterns of circulation affectation. It is also remarkable that the *Group 4* (normal coronary indexes) will be used as a control group, allowing normal baseline levels of circulating miRNAs to be determined and correlated with the other study groups. Additionally, an additional group, *Group 5*, will also provide information on coronary spasms.

Therefore, we consider that the precise definition of the type of coronary circulatory dysfunction, by invasive methods, will allow us to establish a robust relationship between ischemia patterns and biomarkers with potential application in clinical practice, such as miRNAs, to be analysed in our protocol.

## Biological assessment of coronary circulation: Potential of microRNAs as biomarkers

miRNAs are small noncoding RNAs that post-transcriptionally regulate gene expression of approximately 60% of all human genes [34], playing a critical role in mechanistic pathways implicated in the response to stress and the control of homeostasis [5]. Extracellular miRNAs have been detected in body fluids in a stable form [35]. These extracellular transcripts have been proposed as mediators of intercellular communication, including at the autocrine, paracrine, and endocrine level. As a consequence, extracellular miRNAs have been described as biological mediators in adaptive responses, as well as in the onset and development of disease states [8].

A wide array of studies has demonstrated that circulating levels of miRNAs can be analysed in blood specimens and quantified through standard techniques already available in clinical laboratories, i.e., qPCR, reflecting pathological states and offer insights into the molecular phenotype of the patient, including many manifestations of ischemic heart disease [5,8,24,34–38]. All these characteristics seem to make miRNAs especially useful when assessing the involvement of the coronary circulation in chronic coronary syndromes.

To date, multiple miRNAs related to chronic coronary syndromes have been identified, being proposed as biomarkers with potential clinical applications in diagnosis and prognosis. However, the miRNA expression shows wide variability in the literature [24,37,38]. To avoid this methodological problem, we will relate the different coronary patterns only to those circulating miRNAs showing consistent results in the different research works, and we will perform specific analyses to take into account the potential variability on miRNA levels between centres, since this is a multicentric study.

For these reasons, we have designed a rigorous extraction, processing and analysis protocol to reduce variations in the miRNA levels not attributable of other external causes to the type of coronary pattern.

Thus, a strict protocol that relates the miRNA profile to the coronary pattern would allow a biological evaluation of the pathophysiological state of the coronary circulation. This work could establish molecular phenotypes related to alterations in the macrovascular and / or microvascular compartment and their consequences in chronic coronary syndromes.

## Conclusion

The results of *Proyecto ANFIBIO* will contribute to a global and comprehensive understanding of the physiological and biological regulation of coronary circulation by evaluating the

relationship between invasive physiological coronary indexes and the levels of circulating miR-NAs. The *Proyecto ANFIBIO* will also identify potential biomarkers for the diagnosis and clinical management of chronic coronary syndromes.

## Supporting information

**S1 Checklist. SPIRIT 2013 checklist: Recommended items to address in a clinical trial protocol and related documents\*.**
(DOC)

**S1 File. List of all investigators of the involved centers.** IRBLleida, Institut de Reçerca Biomèdica de Lleida; TRRM Group, Translational Research in Respiratory Medicine Group.
(DOCX)

**S2 File. Recommendations for the performance of coronary physiology studies.** RFR, resting full-cycle ratio; FFR, fractional flow reserve; CFR, coronary flow reserve; IMR, index of microvascular resistance; Tmn, mean transit time.
(DOCX)

**S3 File. Data to be obtained in the study.** ACE, angiotensin-converting enzyme; ACS, acute coronary syndrome; aGLP-1, agonist glucagon-like peptide 1; Apo B 100, apolipoprotein B 100; ARA II, angiotensin II receptor antagonists; ARNI, angiotensin receptor/neprilysin inhibitor; CABG, coronary artery bypass graft surgery; CFR, coronary flow reserve; CK, creatine kinase; CPAP, continuous positive airway pressure; CRP, C-reactive protein; DVT, deep vein thrombosis; FFR, fractional flow reserve; GFR, glomerular filtration rate; GGT, gamma-glutamyl transferase; GOT/AST, aspartate aminotransferase; GPT/ALT, alanine aminotransferase; HbA1c, glycated haemoglobin; iDPP4, inhibitor of dipeptidyl peptidase 4; IMR, index of microvascular resistance; iSGLT2, inhibitor sodium-glucose cotransporter-2; HDL, high-density lipoprotein; iPCKS9, proprotein convertase subtilisin/kexin type 9 inhibitor; LA, left atrial; LDH, lactate dehydrogenase; LDL, low-density lipoprotein; LV, left ventricle; MV, mitral valve; NYHA, New York Heart Association; NT-proBNP, N-terminal-pro hormone BNP; Pa, aortic pressure; PCI, percutaneous coronary intervention; Pd, distal intracoronary pressure; PE, pulmonary embolism; PT, prothrombin time; PTT, Partial Thromboplastin Time; RA, right atrial; RFR, resting full-cycle ratio; RV, right ventricle; sPAP, systolic pulmonary arterial pressure; SpO2, peripheral oxygen saturation; STEMI, ST-elevation myocardial infarction; TAPSE, tricuspid annular plane systolic excursion; td, telediastolic; Tmn, mean transit time; TSH, thyroid stimulating hormone; V., velocity; WBC, white blood cells.
(DOCX)

**S4 File.**
(DOCX)

**S5 File.**
(DOCX)

## Author Contributions

**Conceptualization:** Lucía Matute-Blanco, Diego Fernández-Rodríguez.

**Data curation:** Lucía Matute-Blanco, Diego Fernández-Rodríguez.

**Formal analysis:** Lucía Matute-Blanco, Diego Fernández-Rodríguez.

**Investigation:** Lucía Matute-Blanco, Diego Fernández-Rodríguez, Juan Casanova-Sandoval, Thalía Belmonte, Kristian Rivera, Marcos Garcia-Guimaraes, Carlos Cortés Villar, Vicente Peral Disdier, Raúl Millán Segovia, Ignacio Barriuso, David de Gonzalo-Calvo, Ferran Barbé, Fernando Worner.

**Methodology:** Lucía Matute-Blanco, Diego Fernández-Rodríguez, Iván D. Benítez.

**Project administration:** Lucía Matute-Blanco, Diego Fernández-Rodríguez.

**Resources:** Lucía Matute-Blanco, Diego Fernández-Rodríguez.

**Software:** Lucía Matute-Blanco, Diego Fernández-Rodríguez.

**Supervision:** Lucía Matute-Blanco, Diego Fernández-Rodríguez.

**Validation:** Lucía Matute-Blanco, Diego Fernández-Rodríguez.

**Visualization:** Lucía Matute-Blanco, Diego Fernández-Rodríguez.

**Writing – original draft:** Lucía Matute-Blanco, Diego Fernández-Rodríguez.

**Writing – review & editing:** Lucía Matute-Blanco, Diego Fernández-Rodríguez, Juan Casanova-Sandoval, Thalía Belmonte, Kristian Rivera, Marcos Garcia-Guimaraes, Carlos Cortés Villar, Vicente Peral Disdier, Raúl Millán Segovia, Ignacio Barriuso, David de Gonzalo-Calvo, Ferran Barbé, Fernando Worner.

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
