## [Decision Letter · Decision Letter 0]

6 Jan 2023

PONE-D-22-28189

Study protocol for the epigenetic characterization of angor pectoris according to the affected coronary compartment: Global and comprehensive assessment of the relationship between invasive coronary physiology and microRNAs.

PLOS ONE

Dear Dr. Fernandez,

Thank you for submitting your manuscript to PLOS ONE. After careful consideration, we feel that it has merit but does not fully meet PLOS ONE’s publication criteria as it currently stands. Therefore, we invite you to submit a revised version of the manuscript that addresses the points raised during the review process.

ACADEMIC EDITOR: All issues raised by expert reviewers are required. Please edit the manuscript point by point.

We look forward to receiving your revised manuscript.

Kind regards,

Vincenzo Lionetti, M.D., PhD

Academic Editor

PLOS ONE

Journal Requirements:

“DdGC (Miguel Servet 2020: CP20/00041) has received financial support from Instituto de Salud Carlos III co-funded by the European Social Fund (ESF)/ “Investing in your future”.”

“No competing interests”

6. We note that the original protocol that you have uploaded as a Supporting Information file contains an institutional logo. As this logo is likely copyrighted, we ask that you please remove it from this file and upload an updated version upon resubmission.

Reviewers' comments:

Reviewer's Responses to Questions

**Comments to the Author**

1. Does the manuscript provide a valid rationale for the proposed study, with clearly identified and justified research questions?

Reviewer #1: Yes

Reviewer #2: Yes

2. Is the protocol technically sound and planned in a manner that will lead to a meaningful outcome and allow testing the stated hypotheses?

Reviewer #1: Partly

Reviewer #2: Yes

3. Is the methodology feasible and described in sufficient detail to allow the work to be replicable?

Reviewer #1: Yes

Reviewer #2: Yes

4. Have the authors described where all data underlying the findings will be made available when the study is complete?

Reviewer #1: Yes

Reviewer #2: Yes

5. Is the manuscript presented in an intelligible fashion and written in standard English?

Reviewer #1: Yes

Reviewer #2: Yes

6. Review Comments to the Author

You may also provide optional suggestions and comments to authors that they might find helpful in planning their study.

Reviewer #1: The manuscript addresses an interesting topic. The research question is clearly stated. Data will be collected from multiple centers. Some comments follow.

1. The descritpion of the study design must be extended. It is rather unclear how the sample is obtained. Up to what I understand, it is not easy to collect a "reasonable" number of patients for each group (please, explain and justify the sample size was not calculated using standard procedures), but I do not really get why 25 subjects per group is a suited number of collected information. References to other studies should be better discussed.

2. I strongly believe that the difference with other approaches is currently missing. Some relevant references are included, but the novelty of the proposed study is a bit swept under the carpet.

3. I cast several doubts about the employed statistical methods.

a) Counfounders may play a role and statistical matching is rather not feasible with the proposed sample size. This aspects is completely overlooked. Please, remark that association/correlation does not imply causation.

b) The study is a multicenter one. Center-specific effects are neglected. Observations collected at a specific site are more similar than observations collected at a another site. Data have a clear hierarchical strucure that should be accounted for in any statistical analyses.

c) It is rather unclear why a two-step procedure (dimensionality reduction first and clustering then) is assumed. It is well known that this is a not efficient procedure. Clustering and dimensionality reduction can be jointly performed, see e.g. https://link.springer.com/article/10.1007/s11222-008-9056-0.

d) Hierarchical clustering is a too basic approach. Finite mixtures, and in general, model-based approaches should be preferred. PCA is valid only if linear associations can be depicted in the (standardized) data, but it is very likely that non-linear correlations arise.

e) It is rather unclear how the listed methods are related to the research questions.

f) Whataver approach, e.g. testing, regression, etc., you will use, please check that the underlying assumptions are met; otherwise, statistical inference is not valid.

Reviewer #2: The manuscript “Study protocol for the epigenetic characterization of angor pectoris according to the affected coronary compartment: Global and comprehensive assessment of the relationship between invasive coronary physiology and microRNAs.” by Lucía Matute-Blanco et al., described the methodological framework and the objectives of a pilot study aimed at evaluating the relationship between different patterns of coronary involvement and circulating miRNA levels in patients with chronic coronary syndrome. This manuscript is well designed and organized. The focus of this manuscript is new and original. However, tables are not included in the final PDF and some improvements are suggested.

1. Introduction. Please, add Ref. after the sentence “To date, the role of miRNAs in coronary function has been independently evaluated for macro and microcirculation”.

2. Material and Methods.

a. Study design: Please, add Ref. for FFR and IMR; rewrite the sentence “Among the secondary hypotheses to be tested are whether cardiovascular risk factors, medical conditions, medications, echocardiographic findings, angiographic findings, and other invasive physiological indexes are co-related with miRNAs expression.”.

b. Patient selection: “This study will include patients with suspected angina referred for coronary angiography and eventual angioplasty.” Which GL will be followed? Tables are not included in the final PDF.

c. Invasive physiological evaluation of the coronary circulation. Again, more references should be added in order to validate the methods suggested for this study (FFR and IMR), including for guidelines, and for suggested cut-off.

d. microRNA profiling. The methods described in this manuscript is not quantitative and allows to obtain just relative results. Thus, please remove “quantification” from the manuscript and replace with “evaluation”.

e. Please, add Ref. after the sentence “A panel of 14 circulating miRNAs has been selected after an extensive review of the literature.”

f. Sample size estimation. A better explanation should be added for the sample size evaluation. Which is the meaning of “Because studies evaluating the association between circulating miRNAs and coronary patterns are currently lacking, the sample size was not calculated using standard procedures.”?

3. Discussion.

a. Please, add Ref. after the sentence “To date, multiple miRNAs related to chronic coronary syndromes have been identified, being proposed as biomarkers with potential clinical applications in diagnosis and prognosis.”

b. The comment related to the methods for miRNAs evaluation should be modified. Again, the method described in this manuscript is the classical semiquantitative real-time PCR. A quantitative determination of miRNAs is possible only using digital PCR.

7. PLOS authors have the option to publish the peer review history of their article (what does this mean?). If published, this will include your full peer review and any attached files.

Reviewer #1: No

Reviewer #2: No

---

## [Author Response · Author response to Decision Letter 0]

15 Feb 2023

PONE-D-22-28189

Reviewer #1: 

The manuscript addresses an interesting topic. The research question is clearly stated. Data will be collected from multiple centers. Some comments follow.

We thank the reviewer for the suggestions, which allows us to improve our manuscript. We have modified the paper according to reviewer comments.

1. The description of the study design must be extended. It is rather unclear how the sample is obtained. Up to what I understand, it is not easy to collect a "reasonable" number of patients for each group (please, explain and justify the sample size was not calculated using standard procedures), but I do not really get why 25 subjects per group is a suited number of collected information. References to other studies should be better discussed.

 We have expanded the explanation about the “Sample collection”, according to the Reviewer´s comment:

 In each patient, before coronary angiography and any drug administration; 6 mL of blood will be drawn for the centralized miRNAs analyses. Samples will be obtained with support by IRBLleida Biobank (B.0000682) and Plataforma Biobancos PT20/00021. Blood samples will be collected in ethylenediaminetetraacetic acid (BD) blood collection tubes by venipuncture and centrifuged to separate plasma (1500 × g for 10 minutes). Plasma supernatant will be immediately aliquoted, frozen, and stored at -80°C until miRNA profiling. (Page 6, paragraph 4 of the revised paper).

Response to Reviewer #1 and #2.

 We have modified the "Sample size estimation" section, clarifying the proportionality of the sample size of our study in comparison with studies that evaluated the relationship of miRNAs with coronary endothelial function in patients undergoing cardiac catheterization. Below, we present the sample sizes in studies about miRNAs with patients undergoing cardiac catheterization.

- Widmer et al. (PLoS One. 2014;9:e109650) included a total of 48 patients (22=normal vs. 26=abnormal) to evaluate microvascular endothelial dysfunction. 

- Mompeón et al. (Sci Rep. 2020;10:5373) evaluated 86 patients (66=Non-STEMI vs. 20=control volunteers) to assess variations in plasma and serum levels of miRNAs. 

- Mompeón et al. (Cells. 2022;11:1823.) included 104 patients (53=Non-STEMI vs, 51=healthy controls) to determine the miRNA fingerprint in patients with acute myocardial infarction without ST-segment elevation. 

In addition, we provide the justification for considering that 25 patients per group is the minimum number to obtain a representative sample from each group, which would provide a statistical power of 95% for the detection of differences in miRNA levels of one standard deviation between two groups.

Because studies evaluating the association between circulating miRNAs and coronary patterns, assessed by invasive physiological methods, are currently lacking, the sample size was not calculated using standard procedures. Patients of Group 4 (normal coronary indexes) will be used as control group to the Groups 1 to 3 until the recruitment of 25 patients per group will be completed, in a similar number to studies that evaluated the relationship of miRNAs with coronary endothelial function in patients undergoing cardiac catheterization (10,28,29), for a total sample of 100 patients. Based on the t test for the differential expression of miRNAs between two groups, accepting an alpha risk of 0.05 in a bilateral test with 25 patients per group, there is a statistical power of 95% to detect a difference of one standard deviation as statistically significant. It should be noted that patients in Group 5 (coronary spasm) may overlap with patients in Groups 3 and 4; therefore, Group 5 has not been considered for sample size estimation. (Page 9, paragraph 2 of the revised paper).

2. I strongly believe that the difference with other approaches is currently missing. Some relevant references are included, but the novelty of the proposed study is a bit swept under the carpet.

 Currently, the determination of coronary indices, by performing an invasive coronary catheterization, is the best diagnostic tool to evaluate the presence of myocardial ischemia.

miRNAs are considered promising biomarkers for ischemic heart disease, and, to date, multiple studies have been performed, evaluating the relationship of circulating miRNA levels with the different types of ischemic heart disease. However, most of them use non-invasive diagnostic tests for the diagnosis of ischemic heart disease in its different variants. The reduced sensitivity and specificity of non-invasive diagnostic tests, compared to invasive diagnostic tests, could limit the usefulness of miRNAs for the diagnosis of ischemic heart disease.

In addition, there is growing interest in the field of coronary microcirculation, since a relevant percentage of patients undergoing cardiac catheterization do not present obstructive coronary lesions.

To our best knowledge, no other study has been performed comprehensively relating biomarkers, such as miRNAs, proteomics or transcriptomics, with myocardial ischemia, derived from macro or microcirculation, evaluated with the best available diagnostic tools (invasive coronary indices during cardiac catheterization). For these reasons, the ANFIBIO Project could improve the understanding of the role of circulant miRNAs on coronary circulation and be of scientific interest. 

To improve the understanding of the study, we have modified the sentence about the objective of the manuscript in the “Introduction” section: 

Therefore, the objective of the current investigation will be to evaluate globally and comprehensively the relationship between different patterns of coronary involvement, evaluated by invasive methods, and circulating miRNA levels. (Page 3, paragraph 4 of the revised paper).

3. I cast several doubts about the employed statistical methods.

a) Counfounders may play a role and statistical matching is rather not feasible with the proposed sample size. This aspect is completely overlooked. Please, remark that association/correlation does not imply causation.

It should be noted that the propensity score will be included in the empirical Bayes models as confounding factor. It will not be used as a matching method. We have tried to clarify this point in the new version of the manuscript in the “Statistical plan analysis”. 

b) The study is a multicenter one. Center-specific effects are neglected. Observations collected at a specific site are more similar than observations collected at an another site. Data have a clear hierarchical strucure that should be accounted for in any statistical analyses.

Indeed, this is a potential bias. Consequently, the variability between the different centres has been included in the new version of the manuscript. In addition, we will perform additional analysis to take into account this potential variability on miRNA levels, e.g., comparison of global mean expression between independent centres. This analysis has been added to the “microRNA profiling” section. 

Statements, highlighting this aspect, were included in the “microRNA profiling” section:

Global mean expression for each sample will be compared between independent centres in order to analyse potential confounding. (Page 8, paragraph 1 of the revised paper).

and in the “Discussion” section:

To avoid this methodological problem, we will relate the different coronary patterns only to those circulating miRNAs, showing consistent results in the different research works, and we will perform specific analyses to take into account the potential variability on miRNA levels between centres, since this is a multicentric study. (Page 13, paragraph 2 of the revised paper).

c) It is rather unclear why a two-step procedure (dimensionality reduction first and clustering then) is assumed. It is well known that this is a not efficient procedure. Clustering and dimensionality reduction can be jointly performed, see e.g. https://link.springer.com/article/10.1007/s11222-008-9056-0.

We thank the Reviewer for this valuable comment. The analysis has been included in the new version of the manuscript.

d) Hierarchical clustering is a too basic approach. Finite mixtures, and in general, model-based approaches should be preferred. PCA is valid only if linear associations can be depicted in the (standardized) data, but it is very likely that non-linear correlations arise.

As state above, we thank the Reviewer for this valuable proposal. Accordingly, the “Statistical analysis plan” section has been modified.

e) It is rather unclear how the listed methods are related to the research questions.

 The main aim of current investigation is to identify specific circulating miRNAs differentially expressed between groups based on the coronary patterns. To do that, use linear models with empirical Bayes statistics will be used to evaluate differences in miRNA levels between groups. In addition, and attending to your proposal, we will also use parsimonious Gaussian mixture models to identify miRNA signatures associated with each coronary pattern. Finally, and attending to our previous experience in the field, we will also use feature selection based on random forest as a supervised method to define this miRNA signatures.

 Attending to Reviewer`s comment, the “Statistical analysis plan” section has been rewritten.

f) Whatever approach, e.g. testing, regression, etc., you will use, please check that the underlying assumptions are met; otherwise, statistical inference is not valid.

 All statistical analysis will be performed by a statistical expert with extensive experience in this type of analysis (Iván D. Benítez). We have invited him to participate in the project. A new statement has been added to the “Statistical plan analysis” section: 

The assumptions of the statistical models used in the analyses will be adequately tested. (Page 10, paragraph 1 of the revised paper).

Below we show the modified “Statistical analysis plan” section with the modifications according to points 3a, 3b, 3c, 3d, 3e and 3f.

- Statistical analysis plan

The characteristics of the study population will be summarized through standard descriptive statistics. Differences in variables between study groups and correlations between continuous variables will be analysed using the appropriate test (parametric or nonparametric). 

Differential expression of the miRNAs between groups will be evaluated using linear models with empirical Bayes statistics (31). Unadjusted and adjusted models will be fitted, considering a propensity score (PS) to adjust by confounding factors. The PS will be defined as the probability of belonging to a given study group and it will be estimated using a logistic regression model that includes cardiovascular risk factors, medical conditions and medications as predictor variables. Unsupervised clustering of patients based on miRNA levels will be performed using parsimonious Gaussian mixture models (32). The number of latent profiles will be determined using the parsimony criteria based on the minimum value of the Bayesian information criterion measure from 0 to 5 latent profiles. Each patient will be assigned to one class according to his or her highest computed probability of membership. The relationship between the latent profiles and study groups will be evaluated using Fisher's exact test. 

In addition, we will use a machine learning process based on random forest to construct miRNAs signatures associated with discrimination of study groups. Variable importance plots will be displayed to illustrate the prediction value of each miRNA. 

The statistical software package R (www.r-project.org) will be used for statistical analyses. The assumptions of the statistical models used in the analyses will be adequately tested. The two-tailed significance level will be set at <0.05.

Reviewer #2: 

The manuscript “Study protocol for the epigenetic characterization of angor pectoris according to the affected coronary compartment: Global and comprehensive assessment of the relationship between invasive coronary physiology and microRNAs.” by Lucía Matute-Blanco et al., described the methodological framework and the objectives of a pilot study aimed at evaluating the relationship between different patterns of coronary involvement and circulating miRNA levels in patients with chronic coronary syndrome. This manuscript is well designed and organized. The focus of this manuscript is new and original. However, tables are not included in the final PDF and some improvements are suggested.

We thank the reviewer for the suggestions, which allows us to improve our manuscript. We have modified the paper according to reviewer comments.

1. Introduction. Please, add Ref. after the sentence “To date, the role of miRNAs in coronary function has been independently evaluated for macro and microcirculation”.

 We have added new references for this issue and reorganized the references of the entire manuscript. 

2. Material and Methods.

a. Study design: Please, add Ref. for FFR and IMR; rewrite the sentence “Among the secondary hypotheses to be tested are whether cardiovascular risk factors, medical conditions, medications, echocardiographic findings, angiographic findings, and other invasive physiological indexes are co-related with miRNAs expression.”.

 We have added references for FFR and IMR (1,11-13). In addition, we have removed the sentence related to secondary hypotheses because it can cause confusion and the included statistical analyses do not serve this purpose.

b. Patient selection: “This study will include patients with suspected angina referred for coronary angiography and eventual angioplasty.” Which GL will be followed? 

This study will include patients with suspected angina referred for coronary angiography and eventual angioplasty, accordingly to the recommendations of the European Society of Cardiology (1,14). (Page 4, paragraph 5 of the revised paper).

c. Tables are not included in the final PDF.

 Tables had initially been included as other documents. Finally, Tables have been included in the final PDF, embedded in the main manuscript, for a better understanding of the research project.

d. Invasive physiological evaluation of the coronary circulation. Again, more references should be added in order to validate the methods suggested for this study (FFR and IMR), including for guidelines, and for suggested cut-off.

We have added references and suggested cut-off values for FFR and IMR (1,11-14).

d. microRNA profiling. The methods described in this manuscript is not quantitative and allows to obtain just relative results. Thus, please remove “quantification” from the manuscript and replace with “evaluation”.

miRNA profiling will be performed according to the protocol of the miRCURY LNA Universal RT microRNA PCR system (Qiagen). (Page 7, paragraph 3 of the revised paper). The term “quantification” has been removed from the new version of the manuscript. 

e. Please, add Ref. after the sentence “A panel of 14 circulating miRNAs has been selected after an extensive review of the literature.”

 We have added the mentioned references in the “microRNA profiling” section:

A panel of 14 circulating miRNAs has been selected after an extensive review of the literature (19-29). (Page 8, paragraph 2 of the revised paper). 

f. Sample size estimation. A better explanation should be added for the sample size evaluation. Which is the meaning of “Because studies evaluating the association between circulating miRNAs and coronary patterns are currently lacking, the sample size was not calculated using standard procedures.”?

 Response to Reviewer #1 and #2.

 We have modified the "Sample size estimation" section, clarifying the proportionality of the sample size of our study in comparison with studies that evaluated the relationship of miRNAs with coronary endothelial function in patients undergoing cardiac catheterization. Below, we present the sample sizes in studies about miRNAs with patients undergoing cardiac catheterization.

- Widmer et al. (PLoS One. 2014;9:e109650) included a total of 48 patients (22=normal vs. 26=abnormal) to evaluate microvascular endothelial dysfunction. 

- Mompeón et al. (Sci Rep. 2020;10:5373) evaluated 86 patients (66=Non-STEMI vs. 20=control volunteers) to assess variations in plasma and serum levels of miRNAs. 

- Mompeón et al. (Cells. 2022;11:1823.) included 104 patients (53=Non-STEMI vs, 51=healthy controls) to determine the miRNA fingerprint in patients with acute myocardial infarction without ST-segment elevation. 

In addition, we provide the justification for considering that 25 patients per group is the minimum number to obtain a representative sample from each group, which would provide a statistical power of 95% for the detection of differences in miRNAS levels of one standard deviation between two groups.

Because studies evaluating the association between circulating miRNAs and coronary patterns, assessed by invasive physiological methods, are currently lacking, the sample size was not calculated using standard procedures. Patients of Group 4 (normal coronary indexes) will be used as control group to the Groups 1 to 3 until the recruitment of 25 patients per group will be completed, in a similar number to studies that evaluated the relationship of miRNAs with coronary endothelial function in patients undergoing cardiac catheterization (10,28,29), for a total sample of 100 patients. Based on the t test for the differential expression of miRNAs between two groups, accepting an alpha risk of 0.05 in a bilateral test with 25 patients per group, there is a statistical power of 95% to detect a difference of one standard deviation as statistically significant. It should be noted that patients in Group 5 (coronary spasm) may overlap with patients in Groups 3 and 4; therefore, Group 5 has not been considered for sample size estimation. (Page 9, paragraph 2 of the revised paper).

3. Discussion.

a. Please, add Ref. after the sentence “To date, multiple miRNAs related to chronic coronary syndromes have been identified, being proposed as biomarkers with potential clinical applications in diagnosis and prognosis.”

We have added the references in the “Discussion” section:

“To date, multiple miRNAs related to chronic coronary syndromes have been identified, being proposed as biomarkers with potential clinical applications in diagnosis and prognosis. However, the miRNA expression shows wide variability in the literature (24,38,39) (Page 13, paragraph 2 of the revised paper).

b. The comment related to the methods for miRNAs evaluation should be modified. Again, the method described in this manuscript is the classical semiquantitative real-time PCR. A quantitative determination of miRNAs is possible only using digital PCR.

“For these reasons, we have designed a rigorous extraction, processing and analysis protocol to reduce variations in the miRNA levels not attributable of other external causes to the type of coronary pattern” (Page 13, paragraph 3 of the revised paper).

---

## [Decision Letter · Decision Letter 1]

2 Mar 2023

Study protocol for the epigenetic characterization of angor pectoris according to the affected coronary compartment: Global and comprehensive assessment of the relationship between invasive coronary physiology and microRNAs.

PONE-D-22-28189R1

Dear Dr. Fernandez,

We’re pleased to inform you that your manuscript has been judged scientifically suitable for publication and will be formally accepted for publication once it meets all outstanding technical requirements.

Kind regards,

Vincenzo Lionetti, M.D., PhD

Academic Editor

PLOS ONE

Additional Editor Comments (optional):

Reviewers' comments:

Reviewer's Responses to Questions

**Comments to the Author**

1. Does the manuscript provide a valid rationale for the proposed study, with clearly identified and justified research questions?

Reviewer #1: Yes

Reviewer #2: Yes

2. Is the protocol technically sound and planned in a manner that will lead to a meaningful outcome and allow testing the stated hypotheses?

Reviewer #1: Yes

Reviewer #2: Yes

3. Is the methodology feasible and described in sufficient detail to allow the work to be replicable?

Reviewer #1: Yes

Reviewer #2: Yes

4. Have the authors described where all data underlying the findings will be made available when the study is complete?

Reviewer #1: Yes

Reviewer #2: Yes

5. Is the manuscript presented in an intelligible fashion and written in standard English?

Reviewer #1: Yes

Reviewer #2: Yes

6. Review Comments to the Author

You may also provide optional suggestions and comments to authors that they might find helpful in planning their study.

Reviewer #1: All my comments are included in this revised version. I feel that the work is publishable as it stands.

Reviewer #2: The manuscript improved following comments of Reviewers and it is now suitable for publication.

7. PLOS authors have the option to publish the peer review history of their article (what does this mean?). If published, this will include your full peer review and any attached files.

Reviewer #1: No

Reviewer #2: No

---

## [Editor Report · Acceptance letter]

9 Mar 2023

PONE-D-22-28189R1 

Study protocol for the epigenetic characterization of angor pectoris according to the affected coronary compartment: Global and comprehensive assessment of the relationship between invasive coronary physiology and microRNAs 

Dear Dr. Fernández-Rodríguez:

I'm pleased to inform you that your manuscript has been deemed suitable for publication in PLOS ONE. Congratulations! Your manuscript is now with our production department. 

Kind regards, 

on behalf of

Prof. Vincenzo Lionetti 

Academic Editor

PLOS ONE